# Interactive Associations between Physical Activity and Sleep Duration in Relation to Adolescent Academic Achievement

**DOI:** 10.3390/ijerph192315604

**Published:** 2022-11-24

**Authors:** Denver M. Y. Brown, Carah Porter, Faith Hamilton, Fernanda Almanza, Christina Narvid, Megan Pish, Diego Arizabalo

**Affiliations:** Department of Psychology, The University of Texas at San Antonio, 1 UTSA Circle, San Antonio, TX 78254, USA

**Keywords:** exercise, high school grades, movement behaviors

## Abstract

Purpose: The present study aimed to examine independent and interactive associations between physical activity and sleep duration with adolescent academic achievement. Methods: This cross-sectional study used data from the 2019 cycle of the US-based Youth Risk Behavior Surveillance System. A total of 13,677 American adolescents in grades 9 through 12 (M_AGE_ = 16.06 ± 1.24 years; 50.9% female) self-reported their sleep and physical activity behavior as well as their grades. Linear regression models fit with cubic splines were computed to capture potential non-linear associations. Results: Findings for the independent effect models revealed significant curvilinear relationships between physical activity and sleep with academic achievement wherein optimal grades were associated with 7–9 h/night of sleep and 5–7 days/week of physical activity. A significant physical activity by sleep interaction was also observed for academic achievement, which demonstrated that the association between sleep duration and academic achievement is not uniform across levels of physical activity engagement, and tradeoffs may exist. Conclusions: Overall, the results help to identify different combinations of physical activity and sleep behavior associated with optimal academic achievement and suggest that a one-size-fits-all approach to physical activity and sleep recommendations may not be adequate for promoting academic achievement during adolescence.

## 1. Introduction

Inadequate amounts of sleep and physical activity have each been shown independently to have downstream effects on academic achievement [1,2,3,4,5]. Although the exact mechanisms underlying these relationships remain unknown, it is generally understood that the benefits of both behaviors are conferred through improved cognition. Specifically, sleep is believed to play a key role in memory consolidation by strengthening synaptic connections that were active while awake [6,7]. On the other hand, physical activity engagement is known to promote neurogenesis as well as angiogenesis, which positively influence core cognitive processes such as attention, working memory, response inhibition and task switching [8,9]. Taken together, these cognitive benefits manifest through helping students to better recall content they studied previously and concentrate on the task at hand, ultimately resulting in improved learning and higher grades. 

Despite the considerable interest that both sleep and physical activity have garnered in relation to their potential for improving academic achievement, limited research has considered their potential interactive influence. A shortcoming of existing studies is that they have generally examined this relationship based on whether individuals adhere to the respective sleep and physical activity guidelines [10], which neglects much of the variance in these behaviors by failing to consider the full range of responses. Considering academic achievement in secondary school is tied to numerous economic outcomes [11], it is therefore important that we identify the optimal combination of sleep and physical activity to better inform recommendations that will enable students to realize their full potential.

Academic achievement is an important marker of positive adjustment during adolescence and sets the stage for future educational and occupational opportunities [12]. The most serious consequence of school failure, particularly dropping out of school, is the high risk of unemployment or underemployment in adulthood that follows. In contrast, high achievement can set the stage for college or future vocational training and opportunities. For instance, academic achievement is a primary determinant of whether a student gains acceptance to a higher education institution due to the competitive nature of college admissions [13,14,15]. Given that college graduation positively predicts future income, academic achievement in high school can set the stage for one’s future economic success by facilitating social mobility [16,17]. Academic achievement is therefore an important area of inquiry as findings can inform the development of interventions that set students up for success during this critical life stage. 

Research has demonstrated several links between the health behaviors that high school students engage in (or do not) and their academic achievement [18]. The recent emergence of the 24-h movement paradigm has helped to garner more attention for the importance of engaging in a healthy balance of physical activity, sedentary behavior, and sleep over the course of a full day for healthy development during childhood and adolescence [19]. Each of these behaviors has been independently linked with academic achievement [1,3], and researchers are just beginning to explore their interactive effects.

Among the 24-h movement behaviors, physical activity has arguably garnered the most attention for its potential impact on academic achievement [20]. Naturally, this may have occurred due to the inclusion of physical education within many curriculums, which promotes physical activity engagement through the school environment [21]. Findings from a systematic review have suggested an increased curricular emphasis on physical education at the expense of other subjects does not hinder overall academic achievement, but rather, is associated with positive albeit small benefits for grades while at the same time having the potential to provide health benefits [22]. Beyond simply focusing on school-based physical activity, a review of 41 systematic reviews and meta-analyses indicated that regular physical activity participation has a beneficial effect on academic achievement among school-aged children and adolescents [23]. However, it is becoming increasingly apparent that physical activity does not occur in isolation and other movement behaviors adolescents engage in over the course of a day can impact their academic achievement too. 

Sleep is fundamentally different from physical activity in many ways including when sleep behavior typically occurs (i.e., outside of school hours). Nevertheless, the body of literature investigating the relationship between sleep and academic achievement is rapidly developing with studies capturing the importance of various facets of sleep behavior: duration, quality, regularity, and timing [3]. It should be noted that the present paper will focus on sleep duration given the time-based focus of existing public health recommendations [19,24]. Despite an earlier systematic review concluding that shorter sleep durations are negatively associated with academic achievement during adolescence [25], more recent meta-analytic evidence from studies of US adolescents failed to observe a significant relationship (*r* = 0.03) [5]. Beyond these limitations, investigating sleep independently fails to consider that the benefits of regular physical activity engagement may buffer the negative impact of inadequate sleep duration on academic achievement. 

While the interactive influence of adolescent sleep and physical activity behavior has received some attention to date as it relates to mental health outcomes [26,27], academic achievement has yet to be investigated. Research examining 24-h movement guideline adherence lends some insight though. For example, a study of 1290 Spanish adolescents observed the most favorable associations with academic achievement among those who met the physical activity and sleep guidelines; other combinations of guideline adherence (e.g., sleep and screen time, screen time and physical activity)–including concurrent adherence to all three 24-h movement guidelines–demonstrated weaker associations [28]. A key shortcoming of classifying adolescents into groups based on whether or not they adhere to sleep or physical activity guidelines, however, is that these dichotomized groups fail to consider potential dose-response relationships between movement behaviors and academic achievement. Additionally, there has been an overall lack of implementing non-linear approaches within the physical activity and sleep literature that may uncover instances where a “more is better” approach is not accurate, but rather, ranges or different combinations of behaviors may be associated with optimal outcomes. Thus, examining the entire range of values for each behavior and using non-linear modeling techniques stands to help identify an optimal balance of time spent in sleep and physical activity as it relates to academic achievement. 

The present study aimed to investigate the independent and interactive associations of physical activity and sleep with academic achievement among a nationally representative sample of US adolescents. We hypothesized that we would observe: (1) a positive linear relationship between physical activity and academic achievement [29]; (2) an inverted-U relationship between sleep and academic achievement with optimal grades occurring at the public health recommended range of 8–10 h of sleep each night [30]; and (3) based on studies investigating mental health outcomes [27] and mortality risk [31], a physical activity by sleep interaction whereby the beneficial association between sleep duration and academic achievement would be amplified with greater physical activity participation.

## 2. Methods

### 2.1. Study Sample and Data Collection

This cross-sectional observational study used data from the 2019 cycle of the Youth Risk Behavior Surveillance System (YRBSS). The YRBSS is a bi-annual cross-sectional survey that is designed to study many health-related behaviors (e.g., smoking, alcohol, and drug use, diet) in a nationally representative sample of high school students living in the United States. Detailed information about the YRBSS study design, methods, survey measures, and procedures can be found at https://www.cdc.gov/healthyyouth/data/yrbs/index.htm (accessed on 1 February 2022). 

The 2019 cycle of the YRBSS used a multi-stage clustered sampling design to recruit students in grades 9 through 12 (ages 12 to 18 years) in both public and private schools across the United States. A total of 13,677 participants (M_AGE_ = 16.06 ± 1.24 years; 50.9% female) responded to the survey. Participants were sampled from 136 schools within the United States. 

### 2.2. Measures

#### 2.2.1. Sleep

Participants responded to a single item that asked: “On an average school night, how many hours of sleep do you get?” Response options included: “4 or less hours”, “5 h”, “6 h”, “7 h”, “8 h”, “9 h” or “10 h or more”. 

#### 2.2.2. Physical Activity

Participants responded to a single item that asked: “During the past 7 days, on how many days were you physically active for a total of at least 60 min per day? (Add up all the time you spent in any kind of physical activity that increased your heart rate and made you breathe hard some of the time.)”. Response options ranged from 0 to 7 days. 

#### 2.2.3. Academic Achievement

Participants responded to a single item that asked: “During the past 12 months, how would you describe your grades in school?” Response options included: “Not sure”, “None of these grades”, “Mostly Fs”, “Mostly Ds”, “Mostly Cs”, “Mostly Bs”, or “Mostly A’s.” Participants who responded “Not sure” or “None of these grades” were removed from the analysis.

#### 2.2.4. Covariates

Covariates were selected based on established correlations with physical activity [32], sleep [33], and academic achievement [34,35]. These included school grade (9, 10, 11, 12), race/ethnicity (Hispanic, Black, White, Native American, Asian, Mixed), sex (male/female), weight status (normal, overweight, obese), and adherence to screening time guidelines for children and youth (≤2 h per day; yes/no).

### 2.3. Data Analysis

All analyses were performed in R (Version 4.1.1) (R Core Team, Vienna, Austria) and R Studio (Version 2021.09.2) (Posit, Boston, MA, USA). First, we inspected the data for missingness using the *mice* package [36]. Data were considered missing at random and multiple imputations by chained equations was conducted using the *mice* package to replace missing values. A total of 25 multiply imputed datasets were created as per recommendations to set *m* > 100 times the highest fraction of missing information [37]. 

For our primary analyses, a series of linear regression models fit with cubic splines were computed using the *splines* package to examine the independent and interactive effects of sleep and physical activity on academic achievement. Cubic splines were included in our models given that previous work has shown non-linear relationships between sleep duration and academic achievement [38] and very few studies have used non-linear approaches to investigate links between physical activity and academic achievement. A total of six and five knots were included in our physical activity and sleep models, respectively. Visual inspection of academic achievement values revealed a left-skewed distribution and therefore a Gamma distribution was used for all regression models. All models were adjusted for grade, race/ethnicity, sex, weight status, and screen time guideline adherence. Given that beta coefficients for cubic spline models are uninterpretable, a smoothing spline ANOVA model was computed for our interaction effect to determine statistical significance, which was set at α < 0.05. The *survey* package was used to handle the nested structure of the YRBSS dataset (students within schools) [39]. 

## 3. Results

### 3.1. Data Inspection

Missingness ranged from 0.5% for age to 10.9% for weight status (see Table 1). Missingness for physical activity, sleep, and academic achievement was predicted by other observed variables (e.g., more missingness among non-White participants, those with higher body mass index, and those who do not meet the screen time guidelines), which led us to consider data missing at random and use appropriate procedures to preserve our sample size.

### 3.2. Descriptive Statistics

Descriptive statistics for the sample demographic characteristics, physical activity, sleep, and academic achievement are presented in Table 1. 

#### 3.2.1. Physical Activity

Figure 1 shows the relationship between physical activity engagement and academic achievement. Results from our physical activity ANOVA model revealed a significant effect of physical activity on academic achievement (*F*(3,34) = 6.13, *p* < 0.001). 

#### 3.2.2. Sleep

Figure 2 shows the relationship between sleep duration and academic achievement. Results from our sleep ANOVA model revealed a significant effect of sleep on academic achievement (*F*(3,34) = 20.81, *p* < 0.001). 

#### 3.2.3. Physical Activity by Sleep Interaction

Results from our physical activity by sleep ANOVA model revealed significant main effects of sleep (*F*(3,34) = 20.81, *p* < 0.001) and physical activity (*F*(3,31) = 3.53, *p* < 0.001) as well as a significant sleep by physical activity interaction (*F*(9,22) = 2.56, *p* = 0.042). Figure 3 shows the relationship between sleep and academic achievement by the level of physical activity engagement. 

## 4. Discussion

The present study represents the first to examine independent and interactive effects of physical activity engagement and sleep duration in relation to adolescent academic achievement. Using a nationally representative US sample, we took a novel approach in that we included cubic splines within our models which allowed us to observe curvilinear relationships that would not have otherwise been found using traditional linear modeling techniques. Findings for the independent effect models revealed optimal academic achievement was associated with ranges of sleep (7–9 h/night) and physical activity (5–7 days/week) that deviated slightly from respective national public health guideline recommendations for adolescent health (i.e., Physical Activity Guidelines for Americans, American Academy of Sleep Medicine). Our interaction model was also significant, with findings suggesting that the association between sleep duration and academic achievement is not uniform across levels of physical activity engagement, and tradeoffs may exist. Overall, our results help to identify different combinations of physical activity and sleep that are associated with optimal academic achievement and recognize that a one-size-fits-all approach may not exist. 

Contrary to our predictions, our findings revealed a curvilinear relationship between physical activity and academic achievement whereby adherence to the physical activity guidelines did not predict optimal academic achievement, but rather, grades appear to plateau with five or more days of physical activity engagement each week. One potential explanation is that the most active adolescents (i.e., those who reported 7 days/week) spend so much time in these pursuits that it takes time away from endeavors that promote better grades (e.g., studying). For instance, this may be the case for student-athletes who juggle academic responsibilities alongside daily training, practice, and/or competition. Future work should consider using isotemporal substitution modeling to better understand the impact on academic achievement of replacing physical activity with studying (and vice versa); it may be possible that the effects dissipate with increasing levels of physical activity.

Although our finding of a curvilinear relationship between physical activity and academic achievement is in contrast to a previous study that observed a linear relationship despite testing non-linear terms [40], closer inspection of our results suggest relatively linear improvements in academic achievement are observed between one to five days of weekly physical activity engagement. Moreover, results from both studies, and the overarching body of literature indicates that a physically inactive lifestyle or engaging in very low amounts of physical activity is linked to the poorest levels of academic achievement [23]. Taken together, existing evidence provides partial support for the notion that “every move counts” [41], but physical activity promotion efforts should recognize that guideline adherence may be unnecessary for optimal benefits depending on the outcome of interest—academic achievement in this case. Light-intensity physical activity—which does not have a threshold-based public health recommendation—has been shown to positively influence learning during adolescence [42] and therefore could in turn provide additional benefits for academic achievement, but lower intensities of physical activity engagement were not captured by the YRBSS survey. Nevertheless, setting lower physical activity targets such as accruing at least 60 min of moderate-to-vigorous physical activity on five out of seven days of the week could address the knowledge translation issue of an all-or-nothing threshold-based approach and provide many adolescents with the knowledge that they can act upon to improve their grades [43]. 

Akin to our findings for physical activity, optimal academic achievement was associated with amounts of sleep that deviated slightly from public health recommendations. Our hypothesis was partially supported in that we found an inverse-U relationship in which optimal scores for academic achievement were reported by adolescents accruing seven to nine hours of sleep each night as opposed to eight to ten hours as per our predictions based on the American Academy of Sleep Medicine recommendations [44]. The highest grades were observed among adolescents meeting the lower bound of the sleep recommendations (i.e., 8 h). However, sleeping one hour less than recommended (i.e., 7 h) was associated with slightly higher grades than having nine hours of sleep. This is surprising given that research has shown insufficient sleep—among other aspects of sleep such as quality—is associated with poorer learning and school performance [4], and that sacrificing sleep for studying can have negative academic consequences [45]. It was also notable that adolescents who reported sleeping for ten or more hours each night also reported comparable grades to those who slept only five hours. One plausible reason for this may be due to the limited range of response options in the YRBSS; ten or more hours is the highest possible choice and as a result, adolescents who met the upper bound of the sleep recommendations were intermixed with those who exceeded it. Given that empirical evidence has established links between long sleep duration and depression [46,47,48] as well as depression and poorer academic achievement [49,50,51], our findings for ten or more hours should be interpreted with caution. Considered in light of this limitation, our findings suggest getting slightly less than the recommended amount of sleep may not be a detriment to academic performance, but rather, too much sleep is of greater concern. 

Findings from the present study are also the first to demonstrate that physical activity and sleep interact to influence academic achievement. These results suggest potential trade-offs exist in that optimal academic achievement was observed with 7–8 h of sleep duration alongside 5–7 days of weekly physical activity participation, whereas for lower amounts of physical activity (i.e., 1–2 days/week), grades were highest among adolescents accruing at least nine h of sleep each night. Outside of these optimal ranges, however, we found that any level of physical activity cannot buffer the detrimental effects of low amounts of sleep or too much sleep. Similarly, even at optimal amounts of sleep, low amounts of physical activity are associated with poor grades compared to those engaging in greater amounts of physical activity. Collectively, this evidence further underscores the fact that focusing on one movement behavior is short-sighted, and thus adopting an integrative approach that considers multiple movement behaviors may help provide better insight into the range of amounts that may contribute to optimal academic achievement. It should be acknowledged, however, that while engaging in amounts of physical activity and sleep that deviate slightly from public health recommendations may not have a discernable impact on academic achievement, doing so may limit the variety of benefits for healthy development known to be associated with meeting these guidelines during adolescence [19]. 

From a practical application standpoint, schools could consider offering programs that involve frequent tracking of students’ physical activity and sleep behavior for the purpose of offering personalized guidance on how they may change their behaviors to optimize their academic performance. Using self-reported measures may be a cost-effect method to monitor students’ behaviors, although the ubiquity of consumer-grade wearable devices in society today could be leveraged to provide more precise insight into how much physical activity and sleep students are accruing. While the present study used data from the 2019 cycle of the YRBSS, the onset of the COVID-19 pandemic in early 2020 has had a significant negative impact on American students’ academic achievement in the time since [52,53]. In light of our findings, it is plausible that the declines in physical activity behavior [54] and various indices of sleep [55] attributable to the pandemic may partially explain why academic achievement has declined in recent years. Taken together, tailored programming that seeks to improve physical activity and sleep behaviors may be more relevant now than ever. 

Despite several strengths such as using multiple imputations to handle missing data, generalizability through the use of a nationally representative sample, and employing non-linear modeling techniques, this study has limitations that should be acknowledged. First, sleep and physical activity behavior as well as grades were self-reported and as a result, may be biased or prone to recall errors [56]. Second, the physical activity measure employed–while valid for large-scale behavioral surveillance studies–does not assess or capture information about whether physical activity engagement is at a moderate or vigorous intensity, which assumes one hour of moderate-intensity physical activity such as brisk walking is equivalent to one hour of vigorous-intensity physical activity such as sport participation. The instrument also fails to consider physical activities of lighter intensity, which may provide benefits for academic achievement [42]. Future research using instruments that offer more granularity would help to better understand the nature of this relationship. Third, our sleep measure only considered sleep duration, which is but one facet of sleep behavior among many linked to academic achievement. Third, our study employed a cross-sectional design which limits causal inference. Finally, academic achievement is influenced by a myriad of factors including, but not limited to, gender, socioeconomic status, parental involvement, and substance abuse [57,58,59,60].

## 5. Conclusions

Overall, the findings from this study highlight the importance of moving beyond traditional linear approaches to understanding independent and joint associations between physical activity and sleep duration with academic achievement. Doing so helped to recognize that adherence to physical activity and sleep-specific public health guidelines may not be necessary for optimal academic achievement, but rather, amounts slightly below the recommendations may be sufficient. Moving forward, school officials should consider knowledge mobilization efforts that will empower students to engage in greater amounts of physical activity and sleep knowing that they do not need to meet the guidelines to see benefits for their grades.

## Figures and Tables

**Figure 1 ijerph-19-15604-f001:**
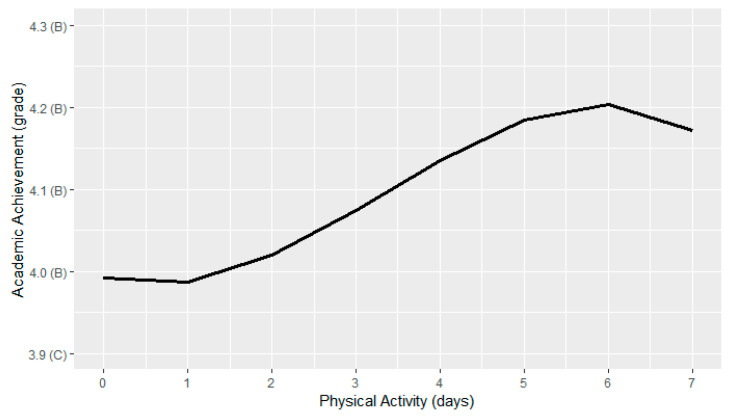
Relationship between physical activity and academic achievement.

**Figure 2 ijerph-19-15604-f002:**
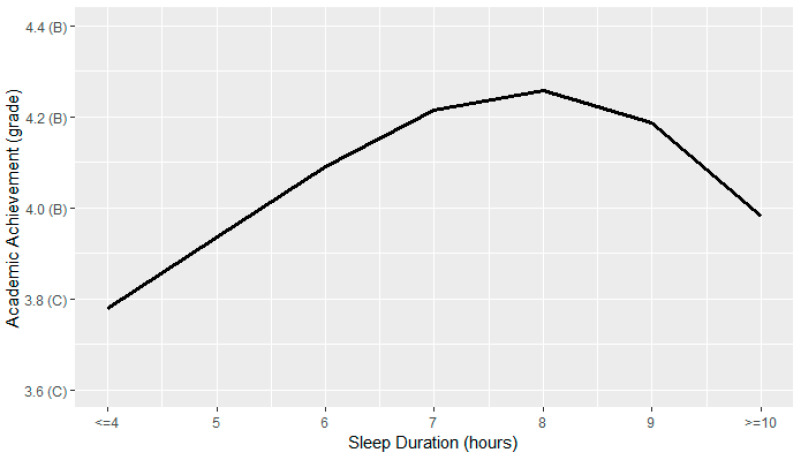
Relationship between sleep duration and academic achievement.

**Figure 3 ijerph-19-15604-f003:**
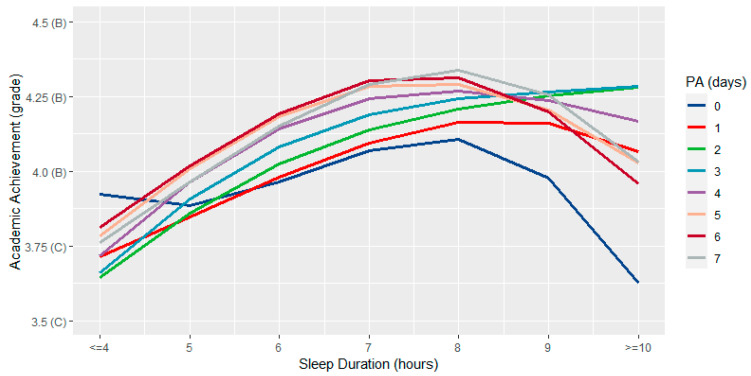
Interactive association between physical activity and sleep in relation to academic achievement.

**Table 1 ijerph-19-15604-t001:** Descriptive statistics for sample demographic characteristics, physical activity, sleep, and academic achievement.

	*N* (%)	Missing
Sex (Female)	6892 (50.9%)	141 (1.0%)
Age		71 (0.5%)
≤14	1760 (13.0%)	
15	3470 (25.6%)	
16	3614 (26.7%)	
17	3099 (22.9%)	
≥18	1594 (11.8%)	
Grade		109 (0.8%)
9	3641 (26.9%)	
10	3719 (27.5%)	
11	3329 (24.6%)	
12	2848 (21.0%)	
Race/Ethnicity		429 (3.1%)
Hispanic	1043 (7.7%)	
White	6794 (50.2%)	
Black	2096 (15.5%)	
Asian	633 (4.7%)	
Native American	221 (1.6%)	
Multi-racial	2749 (20.3%)	
Weight Status		1497 (10.9%)
Normal	9350 (69.1%)	
Overweight	2164 (16.0%)	
Obese	2023 (14.9%)	
Screen Time Guideline Adherence (Yes)	4537 (33.5%)	947 (6.9%)
Physical Activity (days/week)		455 (3.3%)
0	2362 (17.4%)	
1	1039 (7.7%)	
2	1356 (10.0%)	
3	1671 (12.3%)	
4	1344 (9.9%)	
5	1800 (13.3%)	
6	939 (6.9%)	
7	3027 (22.4%)	
Sleep Duration (hours/night)		567 (4.1%)
≤4	1405 (10.4%)	
5	1989 (14.7%)	
6	3372 (24.9%)	
7	3748 (27.7%)	
8	2299 (17.0%)	
9	533 (3.9%)	
≥10	192 (1.4%)	
Academic Achievement		1106 (8.0%)
Mostly As	5500 (40.6%)	
Mostly Bs	5080 (37.5%)	
Mostly Cs	2286 (16.9%)	
Mostly Ds	482 (3.6%)	
Mostly Fs	189 (1.4%)	

Note: Values represent the pooled results for the multiply imputed datasets (*N* = 13,677).

## Data Availability

A publicly available dataset was analyzed in this study. This data can be found here: https://www.cdc.gov/healthyyouth/data/yrbs/data.htm, accessed on 1 February 2022.

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
