# Peer review of "Interactive Associations between Physical Activity and Sleep Duration in Relation to Adolescent Academic Achievement"

_ijerph, 2022, doi:10.3390/ijerph192315604_

Round 1
Reviewer 1 Report
Introduction
In describing the value of academic achievement, please do not limit the benefits to future academic success and related economic benefits, but also include character development and social-emotional benefits for adolescents.
Methods. For non USA readers like me, the phrase “…within all 50 states and the District of Columbia” is unclear.
Discussion
I believe that setting the results of the present study against the notion of “every move counts” is not adequately stated (lines 235 – 238). In fact, I sense that the present results are in line with this recommendation, that is a student may have some physical activity one or two days per week and this is helpful. The fact that in this study physical activity for more than five days did not add to the beneficial effects may be attributed to student athletes’ intensive training, as correctly argued, but this does not necessarily mean that physical activity of low volume and intensity beyond five days has not an effect. For example, taking a walk at the park may be beneficial. Please rephrase this paragraph to this effect including respective clarifying of the sentence in lines 238 – 241.
Please acknowledge that academic achievement is affected by many factors (and probably much more important) such as for example, parental education, parental socioeconomic status, built and social environment and that although focusing on the effect of specific, albeit isolated variables such as sleep and physical activity, although informative for policy, does not provide for the whole picture.
Please acknowledge that the one-week recall measure for physical activity, although valid and perhaps the only choice for a large scale study like the present one, still have disadvantages such as for example, equating five hours of intensive high intensity sport training with one hour of brisk walking. This may have also contributed to the curvilinear association between physical activity and academic achievement which the authors correctly attribute to intensive training.
Author Response
Reviewer 1:
Comments and Suggestions for Authors
Introduction
In describing the value of academic achievement, please do not limit the benefits to future academic success and related economic benefits, but also include character development and social-emotional benefits for adolescents.
Response: We appreciate the Reviewer’s perspective, however, we were unable to find any evidence that demonstrated academic achievement leads to character development and social-emotional benefits. We did, however, find some work that has shown psychosocial factors predict (and precede) academic achievement (Eisenberg et al., 2009; McLeod et al., 2012; Murphy et al., 2015; Tindle et al., 2022). If the Reviewer could kindly direct us to scientific evidence supporting their statement, we will gladly incorporate it into our manuscript.
Methods. For non USA readers like me, the phrase “…within all 50 states and the District of Columbia” is unclear.
Response: We have revised this sentence to read: “ Participants were sampled from 136 schools within the United States.” (L 144)
Discussion
I believe that setting the results of the present study against the notion of “every move counts” is not adequately stated (lines 235 – 238). In fact, I sense that the present results are in line with this recommendation, that is a student may have some physical activity one or two days per week and this is helpful. The fact that in this study physical activity for more than five days did not add to the beneficial effects may be attributed to student athletes’ intensive training, as correctly argued, but this does not necessarily mean that physical activity of low volume and intensity beyond five days has not an effect. For example, taking a walk at the park may be beneficial. Please rephrase this paragraph to this effect including respective clarifying of the sentence in lines 238 – 241.
Response: We kindly disagree with the Reviewer in that our study is not presented against the notion of “every move count” but rather, provides partial support for this messaging in that a plateau effect is found for academic achievement beginning at 5 days of MVPA or more. We do agree with the Reviewer in that light-intensity physical activity could potentially provide benefits for academic achievement, however, the YRBSS survey only captures MVPA. We have revised this paragraph (L 249-255) as well as the limitations section (L 313-320) to reflect the lack of granularity captured by the YRBSS physical activity item(s) with respect to lower physical activity intensities.
“Light intensity physical activity – which does not have a threshold-based public health recommendation – has been shown to positively influence learning during adolescence (43) and therefore could in turn provide additional benefits for academic achievement, but lower intensities of physical activity engagement are not captured by the YRBSS survey. Nevertheless, setting lower physical activity targets such as accruing at least 60 minutes of moderate-to-vigorous physical activity on five out of seven days of the week.”
“Second, the physical activity measure employed – while valid for large scale behavioral surveillance studies – does not assess capture information about whether physical activity engagement is at a moderate or vigorous intensity, which assumes one hour of moderate-intensity physical activity such as brisk walking is equivalent to one hour of high-intensity physical activity such as sport participation. The instrument also fails to consider physical activity of lighter intensity, which may provide benefits for academic achievement (43). Future research using instruments that offer more granularity would help to better understand the nature of this relationship.”
Please acknowledge that academic achievement is affected by many factors (and probably much more important) such as for example, parental education, parental socioeconomic status, built and social environment and that although focusing on the effect of specific, albeit isolated variables such as sleep and physical activity, although informative for policy, does not provide for the whole picture.
Response: We thank the reviewer for highlighting this oversight and agree that movement behaviors such as sleep and physical activity are but two of many factors that influence academic achievement. We have acknowledged in the limitations section that several other factors are also known to influence academic achievement. (L 322-324)
“Finally, academic achievement is influenced by a myriad of factors including, but not limited to, gender, socioeconomic status, parental involvement and substance abuse (58–61).”
Please acknowledge that the one-week recall measure for physical activity, although valid and perhaps the only choice for a large scale study like the present one, still have disadvantages such as for example, equating five hours of intensive high intensity sport training with one hour of brisk walking. This may have also contributed to the curvilinear association between physical activity and academic achievement which the authors correctly attribute to intensive training.
Response: We agree with the reviewer and have updated the Limitations section to reflect this shortcoming of the physical activity instrument. (L 313-320)
“Second, the physical activity measure employed – while valid for large scale behavioral surveillance studies – does not assess capture information about whether physical activity engagement is at a moderate or vigorous intensity, which assumes one hour of moderate-intensity physical activity such as brisk walking is equivalent to one hour of high-intensity physical activity such as sport participation. The instrument also fails to consider physical activity of lighter intensity, which may provide benefits for academic achievement (43). Future research using instruments that offer more granularity would help to better understand the nature of this relationship.”
Reviewer 2 Report
First of all, thank you for being able to review this interesting contribution. The results highlight the opportunity to consider different combinations of weekly physical activity practice and number of sleep hours associated with better grades in school. Among the strengths of the work there are certainly the large number of the sample, the use of cubic splines in the analysis in order to identify non-linear relationships (as reported by similar previous studies) but curvilinear with the related trade-offs. The setting and analytical approach is convincing, even if some additions would be necessary to raise the qualitative level of the work before proceeding to its publication. Here are my suggestions for the authors.
1) In the introduction, the explanatory arguments of the benefit of physical activity and recovery through sleep on school performance are only hinted at, these passages should be better articulated in the text with further references to recent literature.
2) lines 238-241, the period is not entirely clear on reading. It is recommended to rework it.
3) With reference to both the introduction and the discussion, various repetitions of content are noted. It would be advisable to make a precise selection.
4) In order to extend and update the observations that emerge from the study, it would be appropriate to include (in the introduction or in the discussion) a paragraph on how the Covid period has affected both sports and the quality of sleep of adolescents. Several studies have explicitly considered these aspects also with reference to the negative influence on academic performance. The following works could usefully be cited:
https://doi.org/10.5935/1984-0063.20220025;
https://doi.org/10.3390/ijerph19084689;
https://doi.org/10.5935/1984-0063.20200127;
5) In terms of practical applications, the reference to institutional recommendations could be complemented with the opportunity for more precise and frequent monitoring actions of pupils' actual behaviour and general lifestyle, in order to be able to offer personalised guidance actions appropriate to support the achievement of the best school results for everyone.
Author Response
Reviewer 2
Comments and Suggestions for Authors
First of all, thank you for being able to review this interesting contribution. The results highlight the opportunity to consider different combinations of weekly physical activity practice and number of sleep hours associated with better grades in school. Among the strengths of the work there are certainly the large number of the sample, the use of cubic splines in the analysis in order to identify non-linear relationships (as reported by similar previous studies) but curvilinear with the related trade-offs. The setting and analytical approach is convincing, even if some additions would be necessary to raise the qualitative level of the work before proceeding to its publication. Here are my suggestions for the authors.
Response: We thank the Reviewer for their positive views on our manuscript.
1) In the introduction, the explanatory arguments of the benefit of physical activity and recovery through sleep on school performance are only hinted at, these passages should be better articulated in the text with further references to recent literature.
Response: Although the focus of the present paper is not mechanistic by nature, we agree with the Reviewer and have revised the first paragraph to provide an overview of the mechanisms believed to underly the relationships between sleep and physical activity with academic achievement. (L 45-54)
“Inadequate amounts of sleep and physical activity have each been shown independently to have downstream effects on academic achievement (1–5). Although the exact mechanisms underlying these relationships remain unknown, it is generally understood that the benefits of both behaviors are conferred through improved cognition. Specifically, sleep is believed to play a key role in memory consolidation through strengthening synaptic connections that were active while awake (6,7). On the other hand, physical activity engagement is known to promote neurogenesis as well as angiogenesis, which positively influence core cognitive processes such as attention, working memory, response inhibition and task switching (8,9). Taken together, these cognitive benefits manifest through helping students to better recall content they studied previously and concentrate on the task at hand, ultimately resulting in improved learning and higher grades.”
2) lines 238-241, the period is not entirely clear on reading. It is recommended to rework it.
Response: We have revised this sentence to reflect our findings. (L 253-257)
“Nevertheless, setting lower physical activity targets such as accruing at least 60 minutes of moderate-to-vigorous physical activity on five out of seven days of the week could address the knowledge translation issue of an all-or-nothing threshold-based approach and provide many adolescents with knowledge that they can act upon to improve their grades (44).”
3) With reference to both the introduction and the discussion, various repetitions of content are noted. It would be advisable to make a precise selection.
Response: We have reviewed the Introduction and Discussion sections to identify redundant text. We found one major redundancy and this has been removed from the Discussion section: (L 226-229)
“With the exception of one study (23), all non-experimental research investigating the relationship between physical activity and academic achievement has employed linear modeling approaches that generally assume “more is better”(3). We addressed this shortcoming in the literature by using cubic splines.”
4) In order to extend and update the observations that emerge from the study, it would be appropriate to include (in the introduction or in the discussion) a paragraph on how the Covid period has affected both sports and the quality of sleep of adolescents. Several studies have explicitly considered these aspects also with reference to the negative influence on academic performance. The following works could usefully be cited:
https://doi.org/10.5935/1984-0063.20220025;
https://doi.org/10.3390/ijerph19084689;
https://doi.org/10.5935/1984-0063.20200127;
Response: The Reviewer raises a valid point. While the data for the present study was from before the COVID-19 pandemic (2019), our findings offer some important insight given the declines in academic achievement, physical activity behavior and various indices of sleep that have been observed since the onset of the pandemic. We have added a paragraph in the Discussion to focus on these important points: (L 296-308)
“From a practical application standpoint, schools could consider offering programs that involve frequent tracking of students’ physical activity and sleep behavior for the purpose of offering personalized guidance on how they may change their behaviors to optimize their academic performance. Using self-reported measures may be a cost-effect method to monitor students’ behaviors, although the ubiquity of consumer-grade wearable devices in society today could be leveraged to provide more precise insight into how much physical activity and sleep students are accruing. While the present study used data from the 2019 cycle of the YRBSS, the onset of the COVID-19 pandemic in early 2020 has had a significant negative impact on American students’ academic achievement in the time since (53,54). In light of our findings, it is plausible that the declines in physical activity behavior (55) and various indices of sleep (56) attributable to the pandemic may partially explain why academic achievement has declined in recent years. Taken together, tailored programming that seeks to improve physical activity and sleep behaviors may be more relevant now than ever.”
5) In terms of practical applications, the reference to institutional recommendations could be complemented with the opportunity for more precise and frequent monitoring actions of pupils' actual behaviour and general lifestyle, in order to be able to offer personalised guidance actions appropriate to support the achievement of the best school results for everyone.
Response: The Reviewer raises a great point and we have revised the Discussion to acknowledge this excellent potential practical application. (L 296-302)
“From a practical application standpoint, schools could consider offering programs that involve frequent tracking of students’ physical activity and sleep behavior for the purpose of offering personalized guidance on how they may change their behaviors to optimize their academic performance. Using self-reported measures may be a cost-effect method to monitor students’ behaviors, although the ubiquity of consumer-grade wearable devices in society today could be leveraged to provide more precise insight into how much physical activity and sleep students are accruing.”
Reviewer 3 Report
The article entitled "Interactive associations between physical activity and sleep duration in relation to adolescent academic achievement" deals with a very interesting and topical subject whose results can provide findings and considerations regarding the possible relationships and associations between sleep, physical activity and academic achievement in the adolescent population.
However, as contributions and suggestions for improvement, some aspects are indicated:
In the key words, it is recommended not to repeat terms similar to those that already appear in the title of the paper.
It is suggested that the authors deepen their search for scientific literature to contribute to the Introduction and Discussion of the research. In this sense, the support of authors is somewhat reduced, so it would be necessary to add some citations and complement others in the mentioned parts of the study. It should be attempted that the citations to be added are from the year 2017. For example, in the Introduction add some more study to support the argument from line 45 to 51. Support with research the paragraph between line 59 through 62. From line 62 to 66.
In the idea noted in the paragraph from line 72 to 76, it mentions that there have been studies analyzing sleep with academic performance, but it only mentions one. Deepen.
At the beginning of line 87 mention the authors of the Spanish study conducted.
It would be interesting to offer a more contextualized perspective on the one hand in the country where the study was carried out and a small contribution from the international panorama on the subject. In addition to the recent 24-hour movement, mention any other international action related to the subject, for example, what contributions or indications are offered by the WHO, among others.
Justify with authors the statements in the last paragraph of the introduction.
The objective of the study is well reflected. It is suggested that the expectations be stated as research hypotheses.
In the method section, the following aspects are considered:
Mention some examples of the health behaviors to be studied in the survey used.
Indicate in the main text the mean age of the participants, the percentage of female and male participation, as well as the percentage of public and private schools.
It is suggested that the measures (item 2.2) should not be subdivided, but should be presented in a continuous form. Without so many subsections (not necessary).
Regarding the results:
The covariates are presented descriptively but no association was made with the three main variables. Perhaps some relationship could be made according to the sex or weight status of the participants for the findings of this study or for future studies to be conducted.
Improve the title of Table 1.
Improve the presentation of the results. It is not necessary to use so many subheadings to describe the findings according to the three variables.
Do not use X-type symbols in subheading 3.2.3.
In the Discussion section, remember in the first paragraph that the adolescents are Americans.
In line 207 indicate which organization has taken into account in the recommended public health guidelines.
Add in general throughout the section more support from other similar studies carried out.
Although the findings are striking, it would have been interesting to take into account other variables such as the type of physical activity performed by adolescents. It is not the same to perform physical activity with a focus on leisure as on performance.
Finally, we would like to acknowledge the effort made by the authors to carry out a study on a topical subject and whose results may be of relevance for optimizing academic performance through more accurate considerations regarding the practice of physical activity and hours of rest. In spite of finding limitations, these may leave open lines of future research.
Author Response
Reviewer 3
Comments and Suggestions for Authors
The article entitled "Interactive associations between physical activity and sleep duration in relation to adolescent academic achievement" deals with a very interesting and topical subject whose results can provide findings and considerations regarding the possible relationships and associations between sleep, physical activity and academic achievement in the adolescent population.
Response: We thank the Reviewer for their positive view of our manuscript.
However, as contributions and suggestions for improvement, some aspects are indicated:
In the key words, it is recommended not to repeat terms similar to those that already appear in the title of the paper.
Response: We agree with the Reviewer and have revised the keyword “academic performance” to “high school grades” given that “academic achievement” is in the title. Due to this change, “high school” has been removed as an independent keyword.
It is suggested that the authors deepen their search for scientific literature to contribute to the Introduction and Discussion of the research. In this sense, the support of authors is somewhat reduced, so it would be necessary to add some citations and complement others in the mentioned parts of the study. It should be attempted that the citations to be added are from the year 2017. For example, in the Introduction add some more study to support the argument from line 45 to 51. Support with research the paragraph between line 59 through 62. From line 62 to 66.
Response: We have added multiple citations to support these sections. (L 68-72; 82-92; 96-103)
“For instance, academic achievement is a primary determinant of whether a student gains acceptance to a higher education institution due to the competitive nature of college admissions (14–16). Given that college graduation positively predicts future income, academic achievement in high school can set the stage for one’s future economic success through facilitating social mobility (17,18).”
“Among the 24-hr movement behaviors, physical activity has arguably garnered the most attention for its potential impact on academic achievement (21). Naturally, this may have occurred due to the inclusion of physical education within many curriculums, which promotes physical activity engagement through the school environment (22). Findings from a systematic review have suggested an increased curricular emphasis on physical education at the expense of other subjects does not hinder overall academic achievement, but rather, is associated with positive albeit small benefits for grades while at the same time having the potential to provide health benefits (23). Beyond simply focusing on school-based physical activity, a review of 41 systematic reviews and meta-analyses indicated that regular physical activity participation has a beneficial effect on academic achievement among school-aged children and adolescents (24).”
“Nevertheless, the body of literature investigating the relationship between sleep and academic achievement is rapidly developing with studies capturing the importance of various facets of sleep behavior: duration, quality, regularity and timing (3). It should be noted that the present paper will focus on sleep duration given the time-based focus of existing public health recommendations (20,25). Despite an earlier systematic review concluding that shorter sleep durations are negatively associated with academic achievement during adolescence (26), more recent meta-analytic evidence from studies of US adolescents failed to observe a significant relationship (r = 0.03) (5).”
In the idea noted in the paragraph from line 72 to 76, it mentions that there have been studies analyzing sleep with academic performance, but it only mentions one. Deepen.
Response: The citation provided is not for a single empirical study, but rather a comprehensive review of the literature. We therefore feel that a single citation is appropriate here.
At the beginning of line 87 mention the authors of the Spanish study conducted.
Response: After working to revise this sentence, we feel that adding the names of the authors of the study within the text (instead of as a reference at the end of this sentence) negatively impacts its flow. We hope that citing the paper, and making explicit reference to the geographical location of the sample is sufficient.
It would be interesting to offer a more contextualized perspective on the one hand in the country where the study was carried out and a small contribution from the international panorama on the subject. In addition to the recent 24-hour movement, mention any other international action related to the subject, for example, what contributions or indications are offered by the WHO, among others.
Response: As noted in the manuscript, no studies have investigated potential interactive associations between physical activity and sleep in relation to academic achievement and therefore we are unable to couch this study within a global perspective. With the exception of one meta-analysis (Musshafen et al. 2021), all reviews and meta-analytic evidence presented in the manuscript were drawn from global samples and therefore do for the most part provide a synthesis on what is known about independent links between physical activity and sleep with academic achievement. We respect the Reviewer’s point regarding other international action related to the subject, although only a select number of countries have adopted the 24-hr paradigm. The most recent WHO guidelines are specific to physical activity and sedentary behavior, and do not make mention of sleep. Due to this, we feel that acknowledging the WHO guidelines in the Introduction is beyond the scope of this paper.
Justify with authors the statements in the last paragraph of the introduction.
Response: Assuming the Reviewer meant citations instead of authors, we have provided citations from which our hypotheses were based. (L 126-132)
“We hypothesized that we would observe: 1) a positive linear relationship between physical activity and academic achievement (29); 2) an inverted-U relationship between sleep and academic achievement with optimal grades occurring at the public health recommended range of 8-10 hr of sleep each night (30); and 3) based on studies investigating mental health outcomes (27) and mortality risk (31), a physical activity by sleep interaction whereby the beneficial association between sleep duration and academic achievement would be amplified with greater physical activity participation.”
The objective of the study is well reflected. It is suggested that the expectations be stated as research hypotheses.
Response: Revised. (L 126)
“We hypothesized that we would observe…”
In the method section, the following aspects are considered:
Mention some examples of the health behaviors to be studied in the survey used.
Response: Examples are now provided (L 137)
“The YRBSS is a bi-annual cross-sectional survey that is designed to study many health-related behaviors (e.g., smoking, alcohol and drug use, diet) in a nationally representative sample of high school students living in the United States.”
Indicate in the main text the mean age of the participants, the percentage of female and male participation, as well as the percentage of public and private schools.
Response: 1) The YRBSS includes 12 years and younger as well as 18 years or older as response options for the Age item, and therefore, we are unable to provide a mean age of participants. 2) We have included the percentage of females in the main text now (L 143). 3) Information pertaining to the percentage of public and private schools included in the YRBSS is not available and therefore cannot be reported.
“A total of 13,677 participants (50.9% female) responded to the survey.”
It is suggested that the measures (item 2.2) should not be subdivided, but should be presented in a continuous form. Without so many subsections (not necessary).
Response: This may be a matter of IJERPH’s formatting although we feel that subdividing the measures is helpful for researchers seeking to conduct reviews and/or meta-analyses. We leave this decision to the editorial team.
Regarding the results:
The covariates are presented descriptively but no association was made with the three main variables. Perhaps some relationship could be made according to the sex or weight status of the participants for the findings of this study or for future studies to be conducted.
Response: The covariates were selected based on prior research that has shown significant associations with at least one of the main variables (e.g., He et al., 2019; Sterdt et al., 2014; Sousa-Sa et al., 2021). We have updated the Methods section to reflect this. Given that considerable bodies of literature have demonstrated associations between each of the covariates with the main variables (e.g., boys are more active than girls; students with higher BMI achieve lower grades; older adolescents get less sleep) , we do not feel that presenting these associations would be a worthwhile contribution in the manuscript. Doing so would not open the door to future studies and would simply increase the length of the manuscript. (L 163-164)
“Covariates were selected based on established correlations with physical activity (32), sleep (33), and academic achievement (34,35). These included…”
Improve the title of Table 1.
Response: We have updated the title of Table 1 to read, “Descriptive statistics for sample demographic characteristics, physical activity, sleep and academic achievement.”
Improve the presentation of the results. It is not necessary to use so many subheadings to describe the findings according to the three variables.
Response: Again, we respect the Reviewer’s perspective but this may be a matter of IJERPH’s formatting. Nevertheless, we feel it is important to present the findings for the independent physical activity and sleep models, as well as the interactive model separately.
Do not use X-type symbols in subheading 3.2.3.
Response: This has been revised to read, “Physical Activity by Sleep Interaction”
In the Discussion section, remember in the first paragraph that the adolescents are Americans.
Response: We have revised this section to acknowledge the geographical nature of the sample. (L 214)
“Using a nationally representative US sample, we took a novel approach in that we included cubic splines within our models which allowed us to observe curvilinear relationships that would not have otherwise been found using traditional linear modeling techniques.”
In line 207 indicate which organization has taken into account in the recommended public health guidelines.
Response: We have revised this sentence. (L 219-220)
“Findings for the independent effect models revealed optimal academic achievement was associated with ranges of sleep (7-9 hr/night) and physical activity (5-7 days/week) that deviated slightly from respective national public health guideline recommendations for adolescent health (i.e., Physical Activity Guidelines for Americans, American Academy of Sleep Medicine).”
Add in general throughout the section more support from other similar studies carried out.
Response: We have added additional references to support statements in which only one empirical study was cited (Example 1 below). In other areas, only one reference was provided, however, it was to a systematic review or meta-analysis (Example 2). Finally, for certain statements it was not possible to add additional references as there have been very few studies that have investigated associations between sleep and physical activity with academic achievement using non-linear approaches. For example, there has been only one study to date to use a non-linear approach to examine physical activity and academic achievement (i.e., Hansen et al., 2014).
- “Given that empirical evidence has established links between long sleep duration and depression (46–48) as well as depression and poorer academic achievement (49–51), our findings for ten or more hours should be interpreted with caution.” (L 273-276)
- “Moreover, results from both studies, and the overarching body of literature indicates that a physically inactive lifestyle or engaging in very low amounts of physical activity is linked to the poorest levels of academic achievement (23).” (L 244-246)
Although the findings are striking, it would have been interesting to take into account other variables such as the type of physical activity performed by adolescents. It is not the same to perform physical activity with a focus on leisure as on performance.
Response: We couldn’t agree more with the Reviewer in the that benefits for academic achievement may be dependent on the type of physical activity engaged in. If the NHANES dataset has academic achievement as an outcome, this may be possible as it has a more comprehensive battery of physical activity behaviors. We have added further acknowledgment of the limitations of the physical activity instrument in the limitations. (L 313-320)
“Second, the physical activity measure employed – while valid for large scale behavioral surveillance studies – does not assess capture information about whether physical activity engagement is at a moderate or vigorous intensity, which assumes one hour of moderate-intensity physical activity such as brisk walking is equivalent to one hour of high-intensity physical activity such as sport participation. The instrument also fails to consider physical activity of lighter intensity, which may provide benefits for academic achievement (42). Future research using instruments that offer more granularity would help to better understand the nature of this relationship.”
Finally, we would like to acknowledge the effort made by the authors to carry out a study on a topical subject and whose results may be of relevance for optimizing academic performance through more accurate considerations regarding the practice of physical activity and hours of rest. In spite of finding limitations, these may leave open lines of future research.
Response: We thank the Reviewer for the supportive comments and hope our revisions have improved the manuscript.